# IgM+ and IgT+ B Cell Traffic to the Heart during SAV Infection in Atlantic Salmon

**DOI:** 10.3390/vaccines8030493

**Published:** 2020-08-31

**Authors:** Anne Flore Bakke, Håvard Bjørgen, Erling Olaf Koppang, Petter Frost, Sergey Afanasyev, Preben Boysen, Aleksei Krasnov, Hege Lund

**Affiliations:** 1Faculty of Veterinary Medicine, Norwegian University of Life Sciences (NMBU), Ullevålsveien 72, 0454 Oslo, Norway; anne.flore.bakke@nmbu.no (A.F.B.); havard.bjorgen@nmbu.no (H.B.); erling.o.koppang@nmbu.no (E.O.K.); preben.boysen@nmbu.no (P.B.); hege.lund@nmbu.no (H.L.); 2MSD Animal Health Innovation AD, Thormøhlens Gate 55, 5006 Bergen, Norway; petter.frost@bioteknologiradet.no; 3I. M. Sechenov Institute of Evolutionary Physiology and Biochemistry, Torez 44, Saint-Petersburg 194223, Russia; afanserg@mail.ru; 4Nofima AS, Osloveien 1, 1433 Ås, Norway

**Keywords:** Atlantic salmon, B cells, salmon alphavirus, in situ hybridization, IgM sequencing, transcriptome

## Abstract

B cells of teleost fish differentiate in the head kidney, and spleen, and either remain in the lymphatic organs or move to the blood and peripheral tissues. There is limited knowledge about piscine B cell traffic to sites of vaccination and infection and their functional roles at these sites. In this work, we examined the traffic of B cells in Atlantic salmon challenged with salmonid alphavirus (SAV). In situ hybridization (RNAScope) showed increased numbers of immunoglobin (Ig)M^+^ and IgT^+^ B cells in the heart in response to SAV challenge, with IgM^+^ B cells being most abundant. An increase in IgT^+^ B cells was also evident, indicating a role of IgT^+^ B cells in nonmucosal tissues and systemic viral infections. After infection, B cells were mainly found in the *stratum spongiosum* of the cardiac ventricle, colocalizing with virus-infected myocardial-like cells. From sequencing the variable region of IgM in the main target organ (heart) and comparing it with a major lymphatic organ (the spleen), co-occurrence in antibody repertoires indicated a transfer of B cells from the spleen to the heart, as well as earlier recruitment of B cells to the heart in vaccinated fish compared to those that were unvaccinated. Transcriptome analyses performed at 21 days post-challenge suggested higher expression of multiple mediators of inflammation and lymphocyte-specific genes in unvaccinated compared to vaccinated fish, in parallel with a massive suppression of genes involved in heart contraction, metabolism, and development of tissue. The adaptive responses to SAV in vaccinated salmon appeared to alleviate the disease. Altogether, these results suggest that migration of B cells from lymphatic organs to sites of infection is an important part of the adaptive immune response of Atlantic salmon to SAV.

## 1. Introduction

The adaptive immune system of modern bony fish includes B cells, which participate in specific targeting and elimination of pathogens. Teleost B cells produce immunoglobulins (Ig) of three isotypes, i.e., IgM, IgD, and IgT (the latter being equivalent to IgZ in zebrafish) [1,2,3]. While IgM and IgD isotypes are evolutionary conserved and present in all teleost species, IgT is only found in some teleosts, including Atlantic salmon (*Salmo salar*) [4]. The arrangement of the Ig locus prevents isotype switching, which ensures the exclusive expression of membrane-bound IgM, IgM/IgD, or IgT [1,5].

The most abundant immunoglobulin in Atlantic salmon is tetrameric IgM [6]. IgM is found in serum, indicating a prominent role in systemic immunity, but is also secreted to the mucus of mucous membranes [7,8]. The level of IgT in serum of Atlantic salmon is 100–1000 times lower than the level of IgM [9]. IgT is present as a monomer in serum and as a tetramer in gut mucus. IgT consists of three different subclasses in rainbow trout (*Oncorhynchus mykiss*), which differ between organs at the transcriptional level [10]. The ratio of IgT to IgM in rainbow trout is much higher in mucus, and several studies have suggested a primary role of IgT in mucosal immunity of salmonid fish, especially after infection with parasites [11,12,13]. Pathogen-specific responses of IgT were also shown to be present in the internal organs of rainbow trout, namely, the spleen [1] and kidney [14], after infection with viruses and parasites, respectively.

In teleost fish, B cells are believed to originate from lymphopoietic stem cells in the head kidney, making it analogous to mammalian bone marrow [15]. They mature in the head kidney and migrate to sites of activation, such as the spleen or more posterior parts of the kidney [16]. As there are no lymph nodes in teleost [17,18], the spleen constitutes the main secondary lymphatic organ [18]. In addition to head kidney and spleen, populations of B cells with distinct properties are located in different tissues and organs, including mucosa-associated lymphoid tissues (MALTs) [8]. The B cell populations are highly heterogenous, as seen by differential expression of surface cell markers [19] and genes [20,21]. The presence of B cells in peripheral tissues in response to infection is known, e.g., IgM^+^ B cells are recruited to the peritoneum of rainbow trout after injection of bacterium (*Escherichia coli*), *E. coli*-derived lipopolysaccharide, or viral hemorrhagic septicemia virus [22], with increased observation of IgM^+^ and IgT^+^ cells in the pyloric caeca of orally vaccinated rainbow trout [23] and in the skeletal muscle of rainbow trout [24] and Atlantic salmon [25,26] after DNA vaccination. The lack of lymph nodes suggests that terminal differentiation of B cells can occur at various sites, possibly also at the site of inflammation. At present, little is known about the routes and dynamics of B cell traffic and their functional roles under various conditions, including vaccination and viral infection. The recent development of parallel sequencing of the variable region of immunoglobulins (repertoire sequencing (Rep-seq) or immunoglobulin sequencing (Ig-seq)) has opened up novel research possibilities [27,28]. A unique complementarity-determining region (CDR3) or clonotype marks B cells derived from the same ancestor cell, which allows tracking of cell movement between organs and tissues. Each unique CDR3 sequence is a marker of clonal B cells. The presence of a clone in two different tissues indicates a relatively recent migration from one organ to another, with the largest clonotypes indicating recent expansion. Furthermore, RNAscope [29] markedly improved the accuracy and sensitivity of in situ hybridizations, helping to overcome one of the main obstacles in fish immunology, i.e., the limited selection of specific antibodies targeting key cell markers. This technique enables visualization of transcripts of both immune and virus genes within Atlantic salmon tissues [30,31,32,33].

Salmonid pancreas disease virus (SPDV), also known as salmonid alphavirus (SAV), is the causative agent for pancreas disease (PD) [34,35]. PD is characterized by necrosis and loss of exocrine pancreatic tissue, in addition to necrosis and inflammation of the heart and skeletal muscle [36]. Increased Ig transcripts were previously observed in the hearts of salmon infected with SAV [37], suggesting recruitment of B cells. This makes challenge with SAV an attractive model to investigate B cell traffic and their roles in disease protection. Belonging to the genus *Alphavirus*, SAV is a single-stranded positive-sense RNA virus that genetically clusters into six subtypes. Subtypes 2 and 3 (SAV2 and SAV3) are currently found in farmed salmon in Norway with various geographical distribution [36]. The genome of SAV includes nonstructural and structural proteins, with the different subtypes classified on the basis of both structural and nonstructural gene nucleotide sequence [38,39]. Although SAV infections affect multiple tissues, a tissue tropism for the heart (ventricle) was previously demonstrated [40], and tissue from heart samples is commonly used in routine RT-qPCR testing for the virus. Successful vaccination [41,42,43], studies of the adaptive immune responses to SAV [44], and comparison of Atlantic salmon strains with different resistance levels to infection [45,46] point to an important role of immunoglobulins in the protection against pancreas disease. Of note is that the quantitative trait locus (QTL) of Atlantic salmon with higher resistance to PD includes the B locus of the immunoglobulin heavy chain [47].

The aim of the present study was to describe the spatial and functional B cell distribution in the target organ following SAV infection in Atlantic salmon and to investigate B cell traffic by comparing the presence of clonotypes in a major secondary lymphoid organ (SLO) (the spleen) and a target organ (the heart), as well as to assess ensuing heart function by measuring gene expression changes. We observed that SAV challenge enriched IgM^+^ and IgT^+^ in cardiac compartments and found a clonal B cell relationship indicative of virus-induced traffic from the spleen to the heart. Gene expression analysis seemed to show a strong inflammatory response and compromised cardiac physiology in the infected heart of the unvaccinated control group, which was alleviated in the vaccinated fish.

## 2. Materials and Methods

### 2.1. Animal Study

The experimental fish were Atlantic salmon parr/smolt. The animal study was carried out at Veso Vikan Hatchery and Veso Vikan Research Facility (Namsos, Norway). The fish were acclimatized for a minimum of 1 week and starved for a minimum of 48 h prior to vaccination. A total of 51 fish with a mean weight of 62 g were anesthetized (Metacain, Pharmaq) and marked by Passive Integrated Transponder (PIT) tagging prior to vaccination. Vaccination was carried out by intraperitoneal injection with 0.1 mL Aquavac PD7 (MSD Animal Health). The control group (*n* = 52), hereafter named the unvaccinated, were injected with 0.1 mL sterile 0.9% NaCl. Aquavac PD7 vet is a commercial inactivated, multivalent injection vaccine for the immunization of Atlantic salmon. The active components are two inactivated viral antigens, namely, salmon pancreas disease virus (SPDV) and infectious pancreatic necrosis virus (IPNV), and five inactivated bacterial antigens, namely, *Aeromonas salmonicida* subsp. *salmonicida*, *Vibrio salmonicida*, *Vibrio anguillarum* serotype O1, *Vibrio anguillarum* serotype O2a, and *Moritella viscosa*, alongside the oil-based adjuvant. Fish were subjected to 24 h of light stimulus following vaccination. Vaccinated and unvaccinated fish were kept in the same tank with 12 °C flow through fresh water for 47 days before transfer to the research facility. Following transport and prior to challenge, fish were acclimatized to 25‰ salinity at 12 °C for 14 days. SAV3 challenge was performed according to standard procedures at Veso Vikan research facility using a cohabitation challenge model. SAV was provided by The Norwegian Veterinary Institute (Oslo, Norway), Isolate 4 SAV3 210916, 3 passes CHSE-214, titer 106,0TCID50 [48]. The SAV3 virus was diluted 1:5, and 0.1 mL was injected in each shedder. SAV shedders (*n* = 30) were placed into the challenge tank (approximately 450 L) with the cohabitants (vaccinated, *n* = 51; unvaccinated *n* = 52) at 9 weeks post-vaccination. Samples were taken at 0, 21, 35, and 42 days post-challenge (dpc). Shedders were marked by removing the right maxilla. Tissue samples were collected in tubes with RNAlater^TM^ (Sigma Aldrich, St. Louis, MI, USA), which was kindly provided by PatoGen AS (Ålesund, Norway), and stored for 1 day at 4 °C before storage at −20 °C until analysis. Tissues were fixed in formalin (4% formalin, 0.08 M sodium phosphate, pH 7.0), processed in a Thermo Scientific Excelsior^®^ tissue processor (Thermo Fisher Scientific, Waltham, MA, USA), and embedded in paraffin Histowax using a Tissue-Tek^®^, TEC 5 (Sakura Finetek, Alphen aan den Rijn, The Netherlands) embedding center. The challenge trial was approved by the Norwegian Food Safety Authority (permit number 13160).

### 2.2. In Situ Hybridization

Three individuals from each sampling point were subjected to in situ hybridization targeting IgM, IgT, and SAV transcripts using RNAscope 2.5 HD Assay RED (Advanced Cell Diagnostics (ACD), Newark, CA, USA). RNAscope is an in situ hybridization (ISH) method enabling detection of target RNA within tissue or intact cells with high sensitivity and specificity and low background noise due to specific probe design. This is achieved through the use of multiple short probes containing a tail coupling preamplifier, forming a tree-like structure that binds a fluorescent label to produce a strong signal. Briefly, paraffin-embedded tissue sections (4 µm) of heart and pancreas/pyloric caeca were dewaxed at 60 °C for 90 min in ACD HybEZ II, followed by hydrogen peroxide treatment for 10 min while being incubated at room temperature. Samples were boiled in RNAscope target antigen retrieval reagent for 15 min and each section was incubated with RNAscope protease plus at 40 °C for 15 min in a HybEZ oven. Each section was hybridized with the respective RNAscope target probe for 2 h at room temperature. Custom ZZ RNAscope probes were designed to target IgM, IgT, and SAV transcripts (Table 1). The IgM and IgT probes were designed to detect both secretory and membrane-bound forms and all subvariants of each immunoglobulin. SAV inhabits four nonstructural proteins (nsP) named nsP1–nsP4, respectively. Single-stranded, positive-sense RNA can serve both as genomic and mRNA nucleic acid during replication, making it impossible to distinguish between genomic and mRNA target. The nsP1 ISH probe was designed to cover all SAV subtypes. Fast Red chromogenic substrate was used to detect the signals amplified, following the manufacturer’s instructions. Counterstaining was done with 50% Gill’s hematoxylin solution and mounted with EcoMount (BioCare Medical, Pacheco, CA, USA). Imaging was performed by bright field microscopy (Leica microsystems, LM2500, Wetzlar, Germany).

The head kidney was used as the positive control for the IgM and IgT probes. Material from pre- and post-infection served as positive control material for the SAV probe. A probe targeting peptidylpropyl isomerase B (PPIB) in Atlantic salmon was used as a reference target gene to test RNA integrity. A negative control probe (targeting the DapB gene of *Bacillus subtilis*, universal negative control probe) was used to assess cross-reactivity. Both positive and negative control probes were obtained from the manufacturer.

### 2.3. RNA Isolation

Heart and spleen samples (5–10 mg) were placed in tubes with 400 µL lysis buffer (Qiagen), and 20 µL proteinase K (50 mg/m) was added into each tube. Samples were homogenized in FastPrep 96 (MP Biomedicals, Eschwege, Germany) for 120 s at maximum shaking, then centrifuged and incubated at 37 °C for 30 min. RNA was extracted on Biomek 4000 robot using Agencourt RNAdvance Tissue kit (Qiagen Norway, Oslo, Norway) according to the manufacturer’s instructions. RNA concentration was measured with NanoDrop™ One (Thermo Fisher Scientific, Waltham, MA, USA) and quality was assessed with Agilent Bioanalyzer 2100.

### 2.4. Sequencing of the Variable Region of the IgM Heavy Chain

Samples collected at 0, 21, and 42 days post-challenge (dpc) from vaccinated fish (*n* = 8) with a multivalent vaccine (including inactivated SPDV) and unvaccinated controls (*n* = 8) were included in the immunoglobulin sequencing (Ig-seq) analysis. Synthesis of complementary DNA was primed with an oligonucleotide to the constant region (CH) of Atlantic salmon IgM (TAAAGAGACGGGTGCTGCAG) using SuperScript IV reverse transcriptase (Thermo Fisher Scientific, Waltham, MA USA), according to the manufacturer’s instructions. Libraries were prepared with two PCR reactions. The first PCR amplified the cDNA with a degenerate primer, TCGTCGGCAGCGTCAGATGTGTATAAGAGACAGTGARGACWCWGCWGTGTATTAYTGTG, which aligned to the 3′-end of all Atlantic salmon heavy chain variable (VH) genes, with the primer GTCTCGTGGGCTCGGAGATGTGTATAAGAGACAGGGAACAAAGTCGGAGCAGTTGATGA annealing to the 5′-end of CH. Both primers were complementary to Illumina Nextera adaptors. Reaction mixtures (20 µL) included 10 µL 2x Platinum Hot Start PCR Master Mix (Thermo Fisher Scientific, Waltham, MA, USA), 0.5 µL of each primer (10 pmol/µL), 8 µL of water, and 1 µL of template. The second PCR used Illumina Nextera XT Index Kit v2, and the reaction included 2 µL of each primer and 2 µL of the product of the first PCR. The PCR program included heating for 1 min at 94 °C, amplification for 10 s at 94 °C, 20 s at 53 °C, and 20 s at 72 °C (30 cycles in the first PCR and 9 cycles in the second PCR), and extension for 5 min at 72 °C. DNA concentration was measured with Qubit (Thermo Fisher Scientific, Waltham, MA, USA). Aliquots of the libraries were combined and purified twice with the Qiagen PCR clean-up kit. Sequencing was carried out using a Illumina MiSeq Reagent Kit v3 (150-cycle) (Illumina, Inc., San Diego, CA, USA). The libraries were diluted to 4 nM, and PhiX control was added to reach 0.8 nM. After trimming of Illumina adaptors and primers and removal of low-quality reads, sequences were transferred to a database and translated into amino acids. The frequency of each unique sequence (clonotype) was calculated. This study focused on the traffic of B cells between the spleen and the heart. Sharing of IgM repertoires was assessed by the co-occurrence of the largest clonotypes (leaders). Heart–spleen (HS100) and spleen–heart (SH100) metrics were determined as the 100 largest clonotypes of the heart (spleen) that were also detected in the spleen (heart) at a frequency of >10^−4^. Differences were assessed with ANOVA followed by Tukey’s test (Statistica 13).

### 2.5. Microarrays

Analyses were carried out on heart samples from unvaccinated and vaccinated salmon before challenge (0 dpc) and 21 dpc using Nofima’s Atlantic salmon genome-wide 44 k DNA oligonucleotide microarray Salgeno-2 (GPL28080), with a total of 24 microarrays used. The platform was annotated using the bioinformatic pipeline STARS [49]. Microarrays were manufactured by Agilent Technologies (Santa Clara, CA, USA), and the reagents and equipment were purchased from the same provider. RNA amplification and labeling were performed with a One-Color Quick Amp Labeling Kit and a Gene Expression Hybridization kit was used for fragmentation of the labeled RNA. Total RNA input for each reaction was 500 ng. After overnight hybridization in an oven (17 h, 65 °C, rotation speed 0.01 g), arrays were washed with Gene Expression Wash Buffers 1 and 2 and scanned with an Agilent scanner. Subsequent data analyses were performed with STARS. Global normalization was performed by equalizing the mean intensities of all microarrays. Next, the individual values for each feature were divided by the mean value of all samples, thereby producing expression ratios (ER). The log2-ER values were calculated and normalized with locally weighted nonlinear regression (Lowess). Differential expression between vaccinated salmon and control, SAV-challenged, and intact fish was assessed by using an expression ratio of >1.75-fold and *p* < 0.05 (*t*-test). STARS annotations were used for comparison of the functional groups of genes (mean log2-ER, *t*-test).

## 3. Results

### 3.1. IgM^+^ and IgT^+^ B Cell Transcripts Increased in the Heart after Viral Challenge

In situ hybridization targeting IgM (Figure 1) and IgT (Figure 2) transcripts was performed on heart and pancreas/pyloric caeca samples collected at 0, 35, and 42 days post-challenge (dpc) from unvaccinated fish. Prior to challenge (0 dpc), many IgM-transcribing cells were present in the atrium, while only a few scattered positive cells were detected in the ventricle and pancreas (Figure 1A–C, respectively). At 35 dpc, substantial amounts of IgM-transcribing cells remained present in the atrium (Figure 1D), while a marked increase of IgM transcripts was evident in the ventricle and pancreas (Figure 1E,F, respectively). The highest amounts of IgM-transcribing cells were detected in the atrium and ventricle at 42 dpc (Figure 1G,H, respectively), while in the pancreas, the quantity and distribution of IgM transcripts at 42 dpc (Figure 1I) were similar to 35 dpc (Figure 1F). The distribution of IgM transcripts differed within the compartments of the heart. Most IgM transcripts were detected in the atrium and in the *stratum spongiosum* of the heart ventricle.

Only occasional IgT-transcribing cells were detected in the atrium and ventricle prior to challenge (Figure 2A,B, respectively), while no IgT-transcribing cells were detected in the pancreas/pyloric caeca (Figure 2C). The presence of IgT-transcribing cells was evident in both the atrium and ventricle as well as the pancreas/pyloric caeca at 35 dpc (Figure 2D–F, respectively). A further increase in IgT transcripts was observed in both the atrium and ventricle at 42 dpc (Figure 2G,H), showing a similar distribution within the compartments of the heart as that observed for the IgM transcripts (Figure 1). An increase in IgT-transcribing cells was evident in the pancreas/pyloric caeca at 42 dpc (Figure 2I), but to a lesser degree than what was seen in the heart.

### 3.2. Distribution of SAV Transcripts in Target Organs

To study the distribution of virus replication in target organs after challenge, we performed in situ hybridization for SAV nsP1 transcripts on heart samples from unvaccinated fish collected at 0, 35, and 42 dpc. Additional RT-PCR was performed on heart samples from a total of eight individuals at 21 dpc, confirming all individuals as positive for SAV (data not shown). Using in situ hybridization, no viral transcripts were detectable at 0 dpc (Figure 3A). Samples from 35 dpc showed the highest occurrence of viral transcripts. At this time, a few positive cells were observed in the border between the ventricle and the atrium, but viral transcripts were generally mainly detected in the *stratum spongiosum* of the cardiac ventricle (Figure 3B) and within myocardial-like cells of the heart ventricle (Figure 3C). At 42 dpc, the signal was still only detectable in the heart ventricle, and to a lesser degree than 35 dpc (Figure 3D).

### 3.3. Ig-seq Revealed Movement of B Cells from Lymphoid Tissue to the Heart in Response to Infection

To evaluate the traffic of B cells, we looked for co-occurrence of B cell clonotypes, which appear in different organs following migration of clonally expanded cells. We estimated the co-occurrence of leaders (the 100 largest cardiac clonotypes) in the heart that are also found in the spleen (HS100) and, reciprocally, the leaders from the spleen simultaneously present in the heart (SH100). Presuming that B cell expansion occurs in a major SLO (the spleen), and that B cells move from the SLO toward the infected organ (the heart), we used HS100 to detect spleen-to-heart traffic, with SH100 indicating recently expanded clones recruited to the heart. At 21 dpc, HS100 in vaccinated salmon was double that of the unvaccinated controls, indicating larger accumulated migration (Figure 4A), while the higher SH100 in vaccinated salmon compared to unvaccinated fish indicated a significantly larger fraction of recently expanded splenic clonotypes directed to the heart (Figure 4B). Three weeks later (42 dpc), traffic increased in unvaccinated salmon and both metrics equalized in the unvaccinated and vaccinated groups.

### 3.4. Transcriptome Responses: Immune and Cardiac Functions

Microarray analyses performed on heart samples found minor differences between unvaccinated and vaccinated salmon before challenge (Appendix A). After challenge, infected fish showed transcriptome responses to SAV similar to those reported in previous studies [37,46,47]. The changes in relation to uninfected fish seemed to be significantly greater in unvaccinated than in vaccinated salmon, as seen in the numbers of differentially expressed genes (6626 versus 3595 genes), in the functional groups (Figure 5) and individual genes (Figure 6). The strongest responses to SAV and the highest difference between the groups were observed in virus-responsive genes (VRG), a large set of genes activated in response to viruses, double-stranded RNA, and bacterial DNA [26,50], with their expression reflecting the pathogen load. *Isg15* and *viperin* are characterized by the greatest upregulation in infected tissues. Similar, albeit smaller expression changes were indicated in the functional groups of genes associated with signaling via chemokine and cytokine networks, antigen presentation, inflammation, and adaptive immunity. Unvaccinated fish seemed to show higher expression of a suite of genes known as markers of acute inflammation in Atlantic salmon, including the antibacterial peptide *cathelecidin*, *arginase ii*, and immune effectors, all with different modes of action (Figure 6A). This was paralleled by apparently reduced cardiac expression of genes encoding the components of myofiber and proteins that regulate muscle contraction (*ryanodine receptor 1b*, among others) and supply it with energy (sugar metabolism, mitochondria) (Figure 6B). Decreased levels of transcripts for *globins* and erythrocyte markers, such as *rhag*, indicated reduced blood circulation. Reduced expression of *collagens* and proteins specific for epithelium and endothelium was also suggested in the unvaccinated group.

A number of lymphocyte-specific genes were upregulated only in the control group, including genes involved in activation, signaling, and differentiation of lymphocytes (transcription factor *mafb*, *lymphoid-specific helicase, cd166, cd209*, and *lymphocyte cytosolic protein*), B cells (several *cd22-like* genes, *tnf receptor 11b, pu.1, btk tyrosine protein kinase*, and *pi3k regulatory subunit 6*), and T cells (*cd276, cd226*, and *cd4*) (Figure 6B). Two genes involved in V(D)J recombination [51] (*artemis* and *dna cross-link repair 1c*) were downregulated in unvaccinated fish and showed a slight increase in vaccinated fish.

## 4. Discussion

In the present study, the distribution of IgT^+^ and IgM^+^ B cells within the heart of Atlantic salmon before and after SAV infection was examined using in situ hybridization. Furthermore, sequencing of the IgM variable region (Ig-seq) was used to study B cell traffic between the spleen and heart after viral challenge, while the functional consequences of SAV-induced myocarditis in control and vaccinated salmon were assessed using microarray gene expression analyses.

In situ hybridization showed that, prior to infection, the most abundant B cells in the heart were IgM^+^, which were primarily present in the cardiac atrium and less in the ventricle. A previous work showed only a sparse increase in IgM in the heart 1 to 3 weeks after SAV infection by IHC using a polyclonal antibody to recognize salmon IgM [52]. However, in the present study, using in situ hybridization, we detected a large increase in IgM transcripts from 21 to 42 days post-challenge in both the atrium and the ventricle, suggesting a role of B cells in the immune response against SAV in the heart, either by infiltration or by clonal expansion of B cells in the heart. Only a few IgT transcripts were detected in the heart prior to infection, but an increase was evident after SAV challenge. Thus, while IgT was mainly associated with mucosal immunity in fish, these results, as well as previous reports of IgT^+^ cells in the heart [53] and IgT transcripts in the head kidney [14,54] and spleen [1] of salmonids after infection, indicate a role of IgT+ B cells in nonmucosal tissues and in systemic viral infections in salmonids.

IgM^+^ and IgT^+^ B cells were mainly detected in the *stratum spongiosum* and the transition zone between the *stratum compactum* and *stratum spongiosum* of the heart, in addition to some positive cells in the epicardium. This coincided with the localization of nsP1 transcripts corresponding to salmonid alphavirus (SAV) in the heart. Alphaviruses are known to infect muscle cells in other species [55,56], and it was shown that SAV2 targets muscle stem cells in rainbow trout [57]. In Atlantic salmon, virus particles in myocardial cells were indicated by applying electron microscopy [58]. In the current study, viral transcripts were found mainly within myocardial-like cells in the heart ventricle and, to a lesser degree, in equivalent cells in the atrium, suggesting tropism of SAV to ventricular cells in the heart.

As the name pancreas disease suggests, the pancreas is the first organ to develop lesions following infection with SAV, which were detectable at only 2 weeks post-challenge [34]. However, the pancreatic tissue with associated pyloric caeca was previously shown to be unsuitable for RT-PCR detection of SAV [40] due to low levels of viral RNA. In our study, ISH revealed scattered IgM^+^ cells present in the pancreas prior to infection, but low levels after infection in comparison to the heart. Likewise, the levels of IgT transcripts in the pancreas/pyloric caeca were low, and all intestinal layers and smooth muscle, as well as the endocrine and exocrine pancreas, were negative for SAV nsP1 transcripts. These results confirmed that the pancreas is an unreliable source of detecting SAV presence and the ensuing immune response.

While in situ hybridization documented the distribution of B cells within the infected heart, Ig-seq and transcriptome analyses elucidated their origin and possible roles in the defense against SAV infection. IgM sequencing performed in several previous studies revealed a relatively small overlap of the repertoires of the lymphatic and peripheral tissue repertoires under basal conditions [27,28]. A part of newly emerged clonotypes migrate from the lymphatic organs to the target tissues and can be detected at both sites for a limited period of time. A twofold higher co-occurrence of large clonotypes in the hearts and spleens of vaccinated salmon at 21 dpc reflected enhanced traffic of B cells at this time point. B cell traffic to the target organ should not be confused with the rapid, transient, and nonspecific influx of B cells to the sites of vaccination and infection, which is most likely due to their innate and antigen-presenting immune cell roles [59,60] as chemotactic responders to mediators of inflammation [61]. In this study, due to increased traffic of B cells to SAV-infected hearts, we speculate that the delivery of antibody-producing B cells directly at the infection site may increase the effectiveness of the humoral immune response. In vaccinated fish, recruitment of B cells into the infected heart was stimulated much earlier than in control fish.

Our hypothesis of a protective humoral response role in vaccinated individuals was supported by the transcriptomic profiling. At 21 dpc, signs of cardiac dysfunction were detected in the unvaccinated group, including downregulation of genes involved in muscle contraction and energy metabolism, which is typical for PD [37,62]. Decreased levels of transcripts for *globins* and erythrocyte markers, *collagens*, regulators of differentiation, and proteins specific for epithelium and endothelium suggested a lower ability for tissue repair. The upregulation of *arginase ii* and *neuropeptide y* in combination with the downregulation of *calsequestrin* observed in unvaccinated salmon was proposed as a diagnostic symptom of heart pathology caused by PD [37,62]. The differential expression of lymphocyte-specific genes in the infected heart resembled early responses to vaccination in the head kidneys of Atlantic salmon [63]. It is likely that B cells entering the heart of unvaccinated salmon had not yet reached the developmental stage when the surface receptors (BCR) were fully substituted with the secreted IgM. *Pu1, mafb, pi3k regulatory subunit 6*, and *lymphoid-specific helicase* are essential for the differentiation and survival of lymphocytes, with *btk* considered a master regulator of B cell development [64]. *Cd166, cd209, cd226, fermitin family protein*, and *cytohesin 1b* control adhesion and interactions between lymphocytes and the lymphatic tissue environment. The Atlantic salmon genome includes up to 37 *cd22-like* genes, with 19 exhibiting upregulation in the head kidneys of vaccinated salmon, as well as *tnf receptor 11b* [63]. *Artemis* and *dna cross-link repair 1c* proteins, known to be involved in V(D)J recombination [51], were downregulated in the unvaccinated group, which was also previously observed in the head kidneys and spleens of vaccinated salmon. Increased cardiac expression of genes that control differentiation of lymphocytes in Atlantic salmon was observed in previous studies with PD [37,46], suggesting that the terminal stages of lymphocyte maturation can take place at infection sites. In this work, the higher expression of these genes in the unvaccinated group most likely reflected an earlier developmental stage of the lymphocytes, with the B cells of vaccinated salmon possibly reaching more complete differentiation in the spleen or migrating to the heart at an earlier time point than 21 dpc.

## 5. Conclusions

The results of this work suggest that SAV has a tropism for cardiomyocytes in the ventricles of Atlantic salmon. IgM^+^ and IgT^+^ B cells were recruited to the heart in response to SAV challenge, with IgM^+^ B cells being more abundant than IgT^+^ B cells. However, an increase in IgT^+^ B cells was evident, indicating a role of IgT^+^ B cells in nonmucosal tissues and systemic viral infections in salmonids. The viral challenge substantially increased the fraction of splenic B cells directed to the heart, and previous vaccination accelerated this traffic. Gene expression profiling at 21 dpc suggested suppression of various cardiac functions in the control group, which was relieved in the vaccinated fish. Altogether, B cell trafficking to the heart, with subsequent sustainment of cardiac function and reduction of inflammation, may alleviate the severity of this disease. We conclude that traffic of B cells is an essential part of the adaptive humoral responses to SAV in Atlantic salmon.

## Figures and Tables

**Figure 1 vaccines-08-00493-f001:**
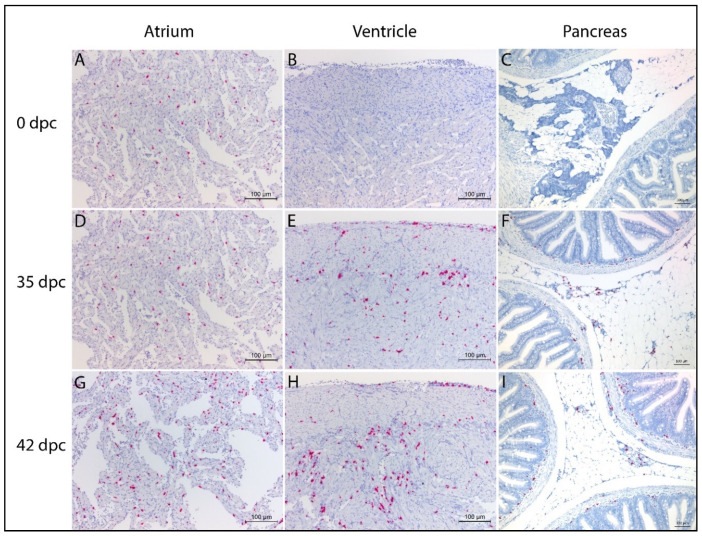
In situ hybridization of immunoglobin (Ig)M transcripts in salmonid alphavirus (SAV)-naïve Atlantic salmon at 0, 35, and 42 days post-SAV challenge (dpc). Images taken from representative individuals within each group. (**A**–**C**) Many IgM-positive cells in the atrium at 0 dpc. Few and scattered positive cells in the ventricle and pancreas. (**D**–**F**) Many IgM-positive cells in the atrium at 35 dpc. Substantial amounts of IgM-positive cells in the ventricle and in the exocrine pancreas. (**G**–**I**) High amounts of IgM-positive cells evident in both the atrium and the ventricle at 42 dpc. IgM-positive cell infiltrates are mainly seen in the *stratum spongiosum* and in the transition zone between the *stratum compactum* and the *stratum spongiosum*. Positive cells also occur in the epicardium. Similar quantities and distributions of IgM-positive cells observed in the pancreas at 42 dpc and 35 dpc.

**Figure 2 vaccines-08-00493-f002:**
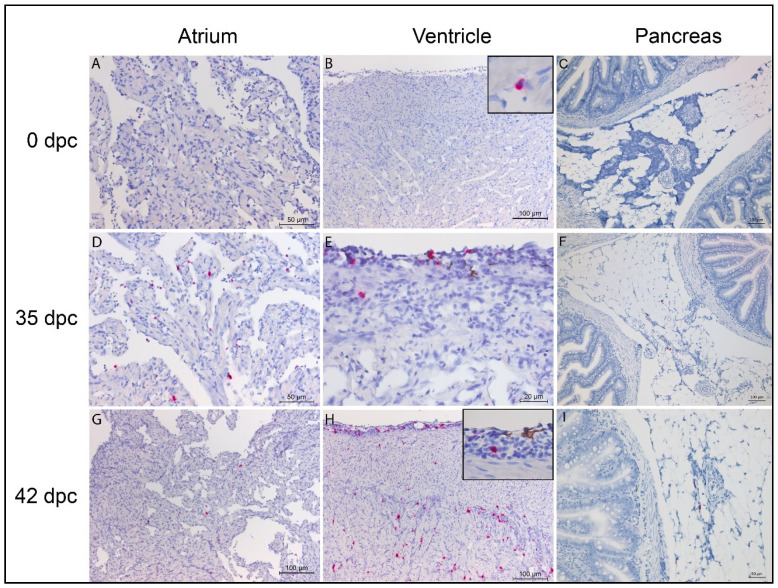
In situ hybridization of IgT transcripts in SAV-naïve Atlantic salmon at 0, 35, and 42 days post-SAV challenge (dpc). Images are taken from representative individuals within each group. (**A**–**C**) Few scattered IgT-positive cells evident in the atrium, ventricle, and pancreas at 0 dpc. (**D**–**F**) Presence of IgT-positive cells in all tissues evident at 35 dpc. (**G**–**I**) IgT-positive cell infiltrates mainly seen in the *stratum spongiosum* and in the transition zone between the *stratum compactum* and the *stratum spongiosum* at 42 dpc. Few scattered IgT-positive cells evident in the atrium and pancreas.

**Figure 3 vaccines-08-00493-f003:**
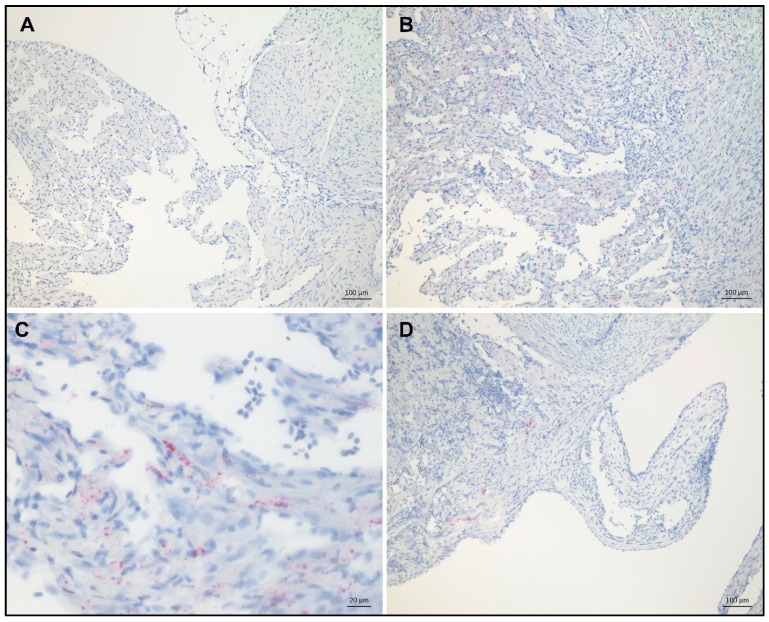
In situ hybridization of nonstructural protein (nsP)1 transcripts in the hearts of SAV-naïve Atlantic salmon at 0, 35, and 42 days post-SAV challenge (dpc). Images taken from representative individuals within each group. Localization of SAV transcripts shown by in situ hybridization of nsP1 in the hearts of unvaccinated fish. (**A**) No positive signal at 0 dpc; (**B**,**C**) virus was mainly detected in the *stratum spongiosum* of the ventricle at 35 dpc; (**D**) virus detected sparingly, mainly in the *stratum spongiosum* of the ventricle, at 42 dpc.

**Figure 4 vaccines-08-00493-f004:**
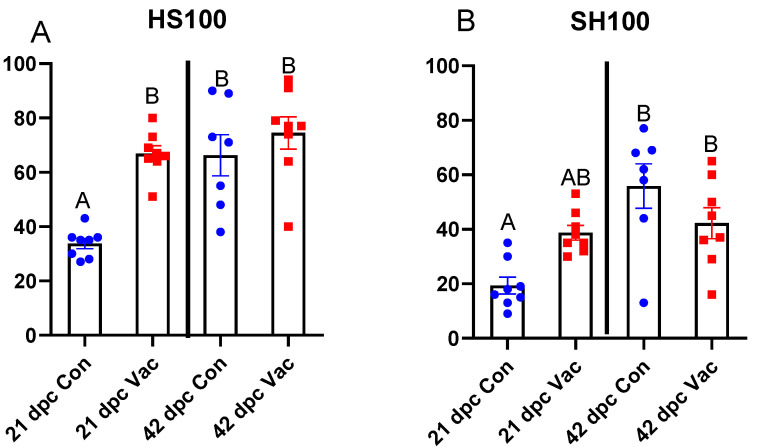
Immunoglobulin sequencing data showing an overlap of repertoires in the heart and spleen of unvaccinated (Con) or vaccinated (Vac) Atlantic salmon 21 and 42 days post-challenge (dpc). (**A**) Occurrence of the 100 most abundant clonotypes in the heart that are also found in the spleen (HS100). (**B**) Occurrence of the 100 most abundant clonotypes in the spleen that are also found in the heart (SH100). Columns not sharing a common letter are significantly different (individual for each figure) (*n* = 8, ANOVA, Tukey’s test, *p* < 0.05).

**Figure 5 vaccines-08-00493-f005:**
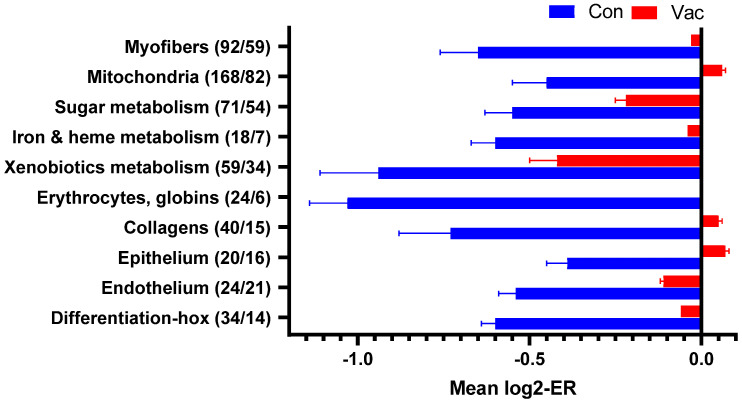
Functional groups of genes with coordinated expression changes in hearts of SAV-challenged Atlantic salmon at 21 dpc in vaccinated fish (vac) versus unvaccinated controls (con). The numbers of differentially expressed genes are in parentheses (saline control/vaccinated). Data are mean log2-expression ratios (infected to intact control) ± Standard Error. All differences between vaccinated fish and saline control are significant.

**Figure 6 vaccines-08-00493-f006:**
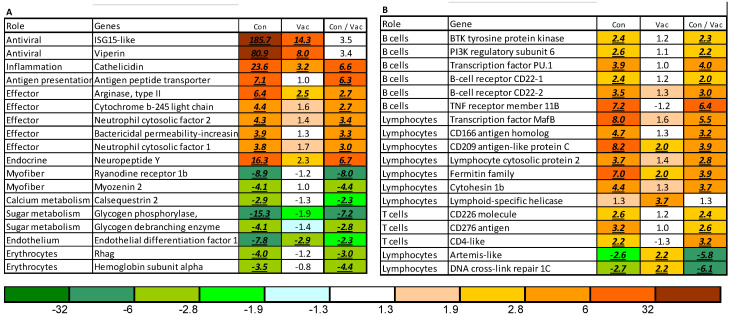
Differentially expressed genes at 21 days post-challenge. Data are the expression ratios (folds) of infected salmon to intact control (first two columns) and saline control to vaccinated fish (third column), highlighted with color scale. Differential expression (>1.75-fold; *p* < 0.05) is indicated with underlined, italic, bold font. (**A**) Inflammation, functional disorders, and heart pathology; (**B**) lymphocyte-specific genes.

**Table 1 vaccines-08-00493-t001:** Target and control probes for in situ hybridization.

	Probe	Accession No.	Target Region (bp)	Catalogue No.
**Target**	IgT	GQ907003	3–883	532171
	IgM	XM_014203125	219–1157	532181
	SAV	NC_003930	410–1279	844631
**Control**	DapB (negative)	EF191515	414–862	310043
	PPIB (positive)	NM_001140870	20–934	494421

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
