# Peer review of "IgM+ and IgT+ B Cell Traffic to the Heart during SAV Infection in Atlantic Salmon"

_vaccines, 2020, doi:10.3390/vaccines8030493_

Round 1

Reviewer 1 Report

In this manuscript, Bakke et al examined traffic of B cells in Atlantic salmon challenged with SAV and found that SAV enriched IgM+ and IgT+ B cell in cardiac compartments. Then they found a clonal B cell relationship indicative of virus-induced traffic from the spleen to the heart. Finally, gene expression analysis showed higher expression of  multiple mediators of inflammation and lymphocyte-specific genes in unvaccinated fish compared to vaccinated, in parallel with massive suppression of genes involved in heart contraction, metabolism and development of tissue. Overall, their study adds new knowledge to the related field, but several concerns still need to be addressed. Comments are listed below:

1. Materials and Methods: Line 119-120, "The SAV3 virus was diluted 1:5 and 0.1 ml was injected in each shedder", what is the viral titer of SAV3 used in this experiment?

2. Figure 1,2,3: For these three figures, it seems images are taken from only one representative individual within each group at each time point. To make the conclusions more convincing, it would be better to choose more representatives and quantify,then compare. Figure 2 legend,In situ hybridization of IgT transcripts in the heart of SAV naïve Atlantic salmon..." is not accurate because this figure is not only about IgT in the heart, also in pancreas. Figure 2 and 3, it would be better to use same magnification for all images.

3. Figure 5: Unvaccinated control is labeled as "S" in Figure 5, please keep all group names consistent throughout the manuscript.

4. Figure 6: What different colors represent for?

5. Page 8 Line 283, "At 21 wpc" should be "At 21 dpc".

Author Response

In this manuscript, Bakke et al examined traffic of B cells in Atlantic salmon challenged with SAV and found that SAV enriched IgM+ and IgT+ B cell in cardiac compartments. Then they found a clonal B cell relationship indicative of virus-induced traffic from the spleen to the heart. Finally, gene expression analysis showed higher expression of  multiple mediators of inflammation and lymphocyte-specific genes in unvaccinated fish compared to vaccinated, in parallel with massive suppression of genes involved in heart contraction, metabolism and development of tissue. Overall, their study adds new knowledge to the related field, but several concerns still need to be addressed. Comments are listed below:

  1. Materials and Methods: Line 119-120, "The SAV3 virus was diluted 1:5 and 0.1 ml was injected in each shedder", what is the viral titer of SAV3 used in this experiment?

Response 1. We thank the reviewer for pointing out this oversight. The viral titer and reference [48] were added.

  1. Figure 1,2,3: For these three figures, it seems images are taken from only one representative individual within each group at each time point. To make the conclusions more convincing, it would be better to choose more representatives and quantify,then compare. Figure 2 legend,“In situ hybridization of IgT transcripts in the heart of SAV naïve Atlantic salmon..." is not accurate because this figure is not only about IgT in the heart, also in pancreas. Figure 2 and 3, it would be better to use same magnification for all images.

Response 2. Three individuals were analyzed at each time-point, and they were all similar in expression of all the transcripts that we tested for. Since they were so similar, we felt it unnecessary to present more than one individual at each time-point. We did not aim to make quantitative estimates, since the differences in the expression of the transcript between the time points were quite obvious, just by looking at the signals in the figures. Thank you for your comment on the figure 2 legend, this has now been edited. With regard to the magnification of images, we sometime used lower magnification to better display distribution of cells within the different layers of the tissue. Higher resolution was used when we wanted to show transcripts within cells. Using the same magnification for all images for figure 2 and 3 would make it difficult to display distribution of transcripts both in different layers off the tissue and within cells.

  1. Figure 5: Unvaccinated control is labeled as "S" in Figure 5, please keep all group names consistent throughout the manuscript.

Response 3. Thank you for this comment. The label was edited and the consistency of names was checked.

  1. Figure 6: What different colors represent for?

Response 4. The color scale was added to the figure.

  1. Page 8 Line 283, "At 21 wpc" should be "At 21 dpc".

Response 5. Thank you, this was edited.

Reviewer 2 Report

Dear Respected authors,

this work builds on previous work that evaluates the immune response in during SAV infection of Atlantic salmon.

the work carries novelty in the microarray study that requires some additional work.

 The authors identified some biological functions related to the activated immune response and altered cardiac functions,

more bioinformatics analysis should be performed on these data, These data should be presented ina heat map to depict the changes among the genes. the way they are described the data is more descriptive rather than a quantitative way.

validation of the genesis a must and I don't think it's available to just rely on the biological, functional assessment to prove that these genes are related to certain functions without validating these genes.

The SAV virus is known to affect the pancreas, the vaccinated vs non vaccinated vs control should be subjected to analysis of the pancreas to show that the infection of the SAV  induced pathological changes.

Minor comments:

the writing of this manuscript seems to be done by different authors where there are certain sections containing challenging sections while others are easy to read.

please revise with a professional English editing specialist.

the authors tend to use acronyms that no one knows what it means:

DPC? QTL? and DEG? 

please describe what is RNAscope®???????????????

Author Response

Dear Respected authors,

this work builds on previous work that evaluates the immune response in during SAV infection of Atlantic salmon.

the work carries novelty in the microarray study that requires some additional work.

 The authors identified some biological functions related to the activated immune response and altered cardiac functions, more bioinformatics analysis should be performed on these data, These data should be presented ina heat map to depict the changes among the genes.

Response 1. Two co-authors (AK and SA) are specialized in microarray research. We use a variety of bioinformatic tools and include in the main text only those results that are informative and consistent with the goals and conclusions. Here, gene expression data are presented as a heatmap with quantitative data: folds with statistically significant differences. Figures like this one have been included in many of our articles; until present, we have published 84 microarray papers.

the way they are described the data is more descriptive rather than a quantitative way. validation of the genesis a must and I don't think it's available to just rely on the biological, functional assessment to prove that these genes are related to certain functions without validating these genes.

Response 2. We have extended description of genes functions and added references.

The SAV virus is known to affect the pancreas, the vaccinated vs non vaccinated vs control should be subjected to analysis of the pancreas to show that the infection of the SAV induced pathological changes.

Response 3. SAV, the causative agent for pancreas disease (PD) in Atlantic salmon, has two target organs – heart and pancreas. This is the reason why we included pancreatic tissue (in addition to heart) when performing ISH. Several other studies report degenerations of the pancreatic tissue shortly after infection with SAV, which is another reason why pancreatic tissue is not used for routine PD testing in Norway. The degeneration of the pancreatic tissue is visible at 35 and 42 dpc in figure 1 and 2. The degenerated pancreatic tissue that we found, in addition to what other research groups have reported on the matter, is a plausible explanation for the few transcripts we detected in the pancreas as compared to the heart. However, pathological changes of pancreatic tissue after infection was not the aim of this study, and we felt that the changes shown in the figures are representative of SAV induced pathology.

 Minor comments:

the writing of this manuscript seems to be done by different authors where there are certain sections containing challenging sections while others are easy to read. Please revise with a professional English editing specialist.

Response 3. Revision used language service from MDPI. Small segments were rewritten to improve the language (highlighted with blue).  

the authors tend to use acronyms that no one knows what it means: DPC? QTL? and DEG? 

Response 4. Thank you for this comment, acronyms were substituted with complete terms.

please describe what is RNAscope®???????????????

Response 5. A short description of RNAscope® was added to Materials and Methods.

Round 2

Reviewer 1 Report

The authors addressed all my comments in the revised manuscript.

Author Response

We are grateful for the review

Reviewer 2 Report

Thank you for the response,

the results presented should be validated using rt-PCR  or Western blotting

Author Response

We are grateful for the review. The manuscript was edited as requested by the Academic Editor to emphasize the indicative character of microarray results.